# How to Manage Conflicts in the Process of ESG Integration? A Case of a Japanese Firm

Alaa Aldowaish [1,*], Jiro Kokuryo [1], Othman Almazyad [2] and Hoe Chin Goi [3]

1 Graduate School of Media and Governance, Shonan Fujisawa Campus, Keio University, Fujisawa 252-0882, Japan; jiro.kokuryo@keio.jp
2 Department of Business Administration, School of Political Science and Economics, Shonan Campus, Tokai University, Hiratsuka 259-1292, Japan; almazyad.othman@tokai.ac.jp
3 NUCB Business School, Nagoya University of Commerce and Business, Nagoya 460-0003, Japan; goi_hc@gsm.nucba.ac.jp
* Correspondence: alaa@keio.jp

**Abstract:** The adoption of environmental, social, and governance (ESG) principles has pressured firms to change their internal operations, creating conflicts regarding meeting sustainability standards. This study uses paradox theory lens to examine conflicts in ESG integration in a business model and explore resolution strategies. We used the case study of a top ESG leader in the information and technology sector—the Ricoh Group. We identified conflicts for environmental, social, and governance factors and found they adopted a synthesis strategy for conflict resolution for all ESG issues. ESG conflicts were resolved by emphasizing that ESG practices are a global requirement with ESG department support and management power. Environmental conflicts were resolved through shifting from cost-of-capital-centric to market-competitiveness-centric. Additionally, social conflicts were resolved through feedback on market needs. We state that using the ESG framework as a promise for future finance, where its adoption of ESG practices may positively impact future financial performance and might support the integration process. We stress the importance of constant feedback with all divisions about ESG regulations and their status and progress toward achieving ESG goals. We further highlight conflict-resolution strategies adopted to support the integration of the ESG framework into the business model.

**Keywords:** ESG integration; conflict management; process; sustainability; business model; paradox theory

## 1. Introduction

Environmental, social, and governance (ESG) reporting has recently shifted from voluntary to mandatory in many countries [1], creating conflicts due to the lack of knowledge about the internal integration of ESG principles [2,3]. Criticisms of the accuracy of ESG reporting in reflecting firms' real sustainability have received increasing attention from various stakeholders, including policymakers, politicians [4], and scholars [5,6].

The majority of ESG discussion in the literature has focused on external aspects of ESG from the financial market [7–14], such as the impact of ESG practices on firms' financial performance, investors' integration of ESG principles into decision-making [15–17], and investors' behavior and rating agencies [18–21].

The internal integration of ESG principles into firm operations is referred to as sustainable development [2], defined in corporate activities as balancing current sustainability with economic, environmental, and social aspects while also addressing company systems, such as operations and production, the organizational system, governance, assessment, and communication [22]. Literature on the internal aspect of ESG practices has focused on corporate governance, such as the impact of board diversity on ESG pursuits [23–25] and ESG disclosure level in reports [26–28].

In terms of conflicts regarding sustainability, the ESG literature discusses the types of conflicts [29–34], proposes strategies to resolve conflicts and tensions [35–37], and finds low reporting of conflicts [38,39]. However, few studies have discussed internal ESG conflicts [40,41], with the discussion limited to the practices of ESG conflict management [40], socio-ecological issues, and financial performance [41]. The literature particularly lacks a conceptual clarity of tensions [33].

This study aims to examine how firms manage conflicts during the ESG integration process using paradox theory. We conducted a case study of a top ESG leader identified from the MSCI Ricoh Group and examined their ESG conflict-resolution strategies. Using a paradoxical lens to understand the nature of these conflicts, we found a synthesis strategy [30,31,42] adopted to resolve conflicts.

We classified the conflicts and resolution strategies based on their category, general ESG conflicts rooted in different perceptions of ESG principles, ways to meet ESG goals, time constraints, and high costs, and resolved them by convincing departments that ESG practices have become a global requirement and providing supports and management power. The environmental conflicts found in the high cost of meeting environmental targets and were resolved by shifting from cost-of-capital-centric to market-competitiveness-centric. Social conflicts in human rights disclosure and increasing percentage of female managers were resolved through negotiation feedback on market needs. Finally, we identified the current issue that causes conflicts is the absence of a well-developed ESG monitoring system.

This study is the first to examine conflicts in the context of the ESG integration process, and the strategies for resolving ESG conflicts. Furthermore, while the majority of the studies examined conflicts from managerial perspectives [42–47], our study focuses on the conflicts that occur in the field. Additionally, we highlight that approaching ESG as a future finance approach may contribute to supporting the ESG integration process.

We propose the following implications for firms: the importance of constant feedback with all their divisions about ESG regulations, and their status and progress toward achieving ESG.

The remainder of this paper is organized as follows: Section 2 discusses the literature on ESG, conflict management and paradox theory. Section 3 presents the research methodology. Section 4 presents and discusses the results. Finally, Section 5 presents the conclusions of this study.

## 2. Literature Review

### 2.1. Environmental, Social, and Governance (ESG)

The ESG framework is originally rooted in sustainability, but includes a broader dimension of sustainability [48], and has emerged to support sustainable finance [49]. According to the recent global sustainable investment review, ESG integration is the second-most adopted tool in 2022, with 5.59 trillion USD in assets under management using this approach [50]. The ESG framework has been considered the best to approach to achieve sustainable development through motivating firms to understand their ESG impact [51] and enhancing competitiveness and reputation [52]. Different notions were formed after the development of the ESG concept, such as the 17 sustainable development goals (SDGs) used in communication with firms and the public [53].

The rating agency market has grown tremendously to support investors' decision-making. Its focus on ESG practices is based on sector priority [54]. However, their methodologies of measuring ESG have been criticized for allowing greenwashing in the manipulation of ESG data [55]. Ref. [56] found no direct relationship between what rating agencies measure and what companies report about their ESG practices.

The literature has extensively examined the impact of ESG practices on firms' financial performance [57], and different findings have emerged. For example, ESG disclosures positively [10,58,59] or negatively affect firms' financial performance [60,61].

By contrast, the literature has examined the internal integration of ESG principles within firms and defined these as "a set of activities or processes associated with an organization's relationship with its ecological surroundings, its coexistence and interaction with human organisms and other populations, and its corporate system of internal controls and procedures (such as processes, customs, policies, laws, rules, and regulations) to direct, administer, and manage all the affairs of the organization to serve the interests of shareholders and other stakeholders" [62]. From this perspective, a rich body of literature has examined ESG disclosure levels [26,27,63–71] and the quality of ESG reporting [28,72,73].

Additionally, the literature has examined the impact of corporate governance sustainability issues on ESG performance [74–80], such as the positive impact of gender diversity and the percentage of women on ESG performance [79,81]. A recent literature review of 49 articles found a positive impact of integrating ESG criteria with corporate sustainability performance, such as improving firm image, competitiveness, and intellectual opportunities [82].

The integration of ESG principles has caused conflicts of interest between different goals, for instance, between management, stakeholders and shareholders [71], financial cost and firms' value [83], with business activities seeking to maximize profit [84], ESG performance, and financial performance [85].

However, few studies have examined conflicts in the process of ESG integration [2], particularly resolving conflicts and tensions to support the integration process. The process of ESG integration is defined as a change, shift, or transition in the business model considering ESG factors in their operations [2].

### 2.2. Conflict Management

Sustainability integration requires the interaction of different stakeholders, which results in tensions and conflicts [36]. Conflict has been defined as the tension between contradicting ideas [86]. Moreover, it is classified into hard and soft: hard occurs between the logic of financial rationality and sustainability, and soft might refer to opinions based on the nature of the conflict.

The literature has identified different types of internal sustainability tensions. Ref. [30] classified these into three groups: strategic direction, domain, and strategy implementation. Ref. [32] classified sustainability tensions into two categories. First, there are non-temporal tensions of business sustainability, such as tensions across and within the triple bottom line, across levels, and within the firm. Second, intertemporal tensions in business sustainability include tensions between real and perceived future needs, anchoring tensions, asynchrony tensions, and other subjective tensions. Additionally, [31] identified four types of tensions: personal versus organizational, sustainability agendas, corporate short- versus long-term orientation, isomorphism versus structural and technological change, and efficiency versus resilience of socioeconomic systems. Ref. [29] identified two types of tensions between short- and long-term goals and external tensions between business and society. Ref. [33], a literature review, examined tensions, paradoxes, dilemmas, and trade-offs in the literature and developed an analytical framework for assessing tensions in sustainability transition. The authors identified six tensions that can happen in intra- and interorganizational contexts. These include tension between private and shared values, personal vs. organizational sustainability agenda, isomorphism vs. structural and technological change, efficiency vs. resilience, further categories of tension, and an unspecific category of tension. Furthermore, Ref. [34] examined tensions in business model innovation for sustainability using a single case study of Australian firms and found that the primary source of tension was perceived power imbalance and conflicts in values between sales forces that focus on social impact and distribution channels that focus on financial outcomes. For instance, the CEO strategy focuses on profitability, and some employees resist this focus. Tensions were also found in the conflict between customer ownership, product ownership, and incentives. Furthermore, the perception of sustainability at the top level differs from that at the middle level [45].

In addition, the literature has examined the types of conflicts managers deal with in sustainability integration. For example, [46] found three tensions that managers encounter in implementing sustainability strategies: tension between product features, values, and goals. Ref. [87] examined how managers deal with tensions in 25 forestry and wood product organizations in Australia. The authors found acknowledgement of sustainability tensions in the sector and different perceptions of tensions, such as standardization and efficiency versus advancing environmental and social practices, short-term financial governance versus long-term environmental and social governance, and core business activities versus local community engagement. The authors classified the tension management strategies based on [31] as acceptance, separation, and synthesis strategies. Ref. [43] examined the sources of managers' tensions, as well as their types, and reactions toward meeting global standards. Voluntary certificates or policies for meeting each of the environment, social, and governance factors also exist, such as the environment certificate ISO 14001 standard in 23 Japanese and 20 South Korean firms. The authors found that managers in both countries encountered societal–commercial, traditional–modern, and individual–collective tensions. Additionally, Ref. [47] examined management sensemaking in sustainability. The authors explained the differences in terms of managerial scanning, interpreting, and responding to sustainability issues depending on whether decision-makers hold a business case frame or paradoxical frame.

The literature proposes methodologies for managing conflicts and tensions. Ref. [35] argued that identifying stakeholders, the impact of change on them, and resistance factors are essential to conflict management in firm change. They proposed a model for managing conflict in organizational change. The model consists of stakeholder identification using a boundary critique, determination of crucial resistance factors, application of network mechanisms and intervention strategies, and evaluation of strategies and outcomes. Furthermore, [36] examined the role of conflict management in building social sustainability through a literature review, with the authors proposing a framework to understand the paradox and reflexibility in moving toward social sustainability. This methodological framework consists of social sustainability and the management of multi-stakeholder processes. The first approach is to provide normative frameworks, such as indices and standardized guidelines. The second approach is related to describing and analyzing practices and case studies of stakeholder management for social sustainability. Managing sustainability conflicts can be done through three strategic approaches: applying common evaluative frameworks, building contextual convergence, and embracing complexity [37].

Organizational culture plays an essential role in managing sustainability conflict. Ref. [88] examined the impact of four culture types (clan, adhocracy, market, and hierarchy) on sustainable innovation performance, and found that clan culture negatively affects sustainability performance and has a positive effect on hierarchy and adhocracy culture. Ref. [89] examined tensions and trade-offs in pursuing the social and financial goals of German social enterprise and found three strategies to deal with tensions: reconciliation strategies, structural and temporal separation, and acceptance strategies. Ref. [30], a literature review of 149 articles, examined how the literature addresses sustainability management tension and classified it into four strategies: win–win, trade-offs, integrative, and paradoxical. Additionally, [90] examined three approaches to respond to sustainability-related legitimacy issues: "one best way," contingency, and paradox.

A few studies have examined ESG conflicts in firms. For example, [40] proposed a methodology to resolve conflict as follows: managing conflict, personal administration, compliance with ethical standards, the culture of declaring conflict, and limiting proprietary information circulations. In [41], it is argued that ESG is best for resolving financial conflicts because firms with a high ESG index are more profitable.

In the literature, different theories have been used to understand conflicts and tensions. For instance, contingency theory is used to examine the conditions of tension resolution, whereas paradox theory examines the tensions simultaneously [91] and considers as a more systematic and holistic way to understand sustainability by exploring new possibilities [30].

*2.3. Paradox Theory*

The paradox view is defined as a paradox perspective on corporate sustainability that accommodates interrelated yet conflicting economic, environmental, and social concerns to achieve superior business contributions to sustainable development [92]. The paradox theory helps in understanding how tensions are managed [29,30]. Moreover, it has been argued to be the best approach to resolving tensions and conflicts [30].

The paradox approach is comprised of three aspects: descriptive, instrumental, and normative. The descriptive aspect describes how firms deal with paradoxical tensions, whereas the instrumental aspect establishes connections between different sustainability tensions and outcomes. The normative aspect concerns the belief that firms are responsible for environmental and social factors that reach beyond financial performance and value in approaching different sustainability goals [92]. Ref. [42] classified paradox tensions of the sustainable business model into four types: performing tension, which emerges from different stakeholders' goals; belonging/identity tension, which emerges from conflicting identity and values; organizing tension, which emerges from the internal dynamics of culture, leadership, and structure; and learning/temporal tension, which emerges from multiple time horizons, such as growth, change, and flexibility. The author proposed three strategies for managing sustainability paradoxical tensions: suppression, acceptance, and resolution strategies. Ref. [93] conducted a systematic literature review of 53 studies and categorized them into three research areas: paradoxical tension, paradoxical frame/thinking, and paradoxical actions/strategies. Ref. [31] proposed an integrative approach to solve tensions in sustainability using paradox-strategy dimensions from three perspectives: level, change, and context.

Limited literature has discussed ESG integration from a paradoxical perspective. The discussion mainly regards investors' behavior [94–96]. For instance, [96] examined the ESG integration paradox from an investor perspective. The ESG paradox existed because of the difficulties in aligning long-term ESG benefits with firms' short-term performance. Moreover, [94] examined paradox behavior that resulted from pressure to meet ESG goals and actual investment behavior. Hence, this study employs paradox theory to discuss the conflict of ESG integration from an organizational behavior perspective.

## 3. Methodology

We employed a case study of a top ESG leader in the information and technology sector identified by MSCI, one of the largest ESG rating agencies [97]. A single case study helps to understand complex and contemporary phenomena in depth [98]. Furthermore, the main research questions involve "how" questions, in which the case study is considered the appropriate method to answer our questions [98]. The information and technology sector is considered to have the third-largest sector weight [99], facing pressure from different stakeholders for ESG disclosures. The Ricoh Group was selected based on the following criteria. First, we identified it from an analysis of corporate narrative reports of all ESG leaders identified from the MSCI rating agency in 2022 of 25 Japanese firms. Ricoh was the only firm that explicitly linked ESG goals with their business model, highlighting the process where it plans to change its business model in transitioning toward a digital service business model. Second, Ricoh was included in the 2021 list with an ESG rating of A, which was changed to AA in the following year. Finally, in our analysis of 25 firms' corporate narrative reports in 2022, we found that Ricoh demonstrated the highest frequency of ESG reports, for a total of 401 ESG keywords.

Furthermore, Ricoh is considered a large firm, and larger firms tend to face more conflict than small firms because of different stakeholders goals and values [100]. Our research strategy is an explorative interpretation [101] used to understand firm processes [102].

Table 1 shows our data-collection methodology, which consists primarily of data sources from semi-structured interviews. Second-hand data sources included document analysis, websites, and press releases. Interviews were conducted between May 2023 and October 2023. Interviews were conducted three times: one was face to face and two were

conducted online. The three interviews lasted 60 min each. We developed a case study protocol based on [98]. After discussing the literature on ESG and conflict management with our research team, we devised a set of questions. In the first interview, we introduced our research and asked general questions about ESG principles. In the second and third interviews, we asked questions about ESG conflicts and how they were addressed. For instance, we started with broad questions to understand how Ricoh perceived the conflicts.

**Table 1.** Data sources.

| Data | Date |
| --- | --- |
| Research background and semi-structured interview<br>• CSV section leader, ESG center business promotion department, professional service division | 6 April 2023 |
| Semi-structured interview<br>• General manager, ESG strategy division, ESG center<br>• CSV section leader, ESG center business promotion department, professional service division | 23 May 2023 |
| Semi-structured interview<br>• General manager, ESG strategy division, ESG center<br>• CSV section leader, ESG center business promotion department, professional service division | 30 October 2023 |
| Private document<br>• Initiatives to integrate ESG/SDGs and management strategy 2023<br>Public documents:<br>• Ricoh Group Integrated Report 2023<br>• RICOH Group ESG Data Book 2022<br>• Ricoh Group Integrated Report 2012- 2022<br>• Ricoh Group TCFD Report 2022<br>• Ricoh Group Circular Economy Report 2022 | 2022–2023 |

Do you follow a specific strategy for ESG conflict resolution?

How do you solve conflicts in pursuing ESG goals?

How do you convince sales managers to adopt ESG practices?

Follow-up questionnaires were included, where we asked for more examples and clarifications.

We requested interviews with the general manager of the ESG strategy division and a CSV section leader, as they are responsible for all the details related to ESG practices. They have also been working at the Ricoh Group for more than ten years and have an in-depth understanding of the changes. Two researchers attended the interview, and all the interviews were audio-recorded and transcribed using MAXQDA 2022 2 software.

A grounded theory strategy approach was adopted [103]. Our data analysis consisted of abductive reasoning. Going through the data back and forth from our empirical findings and theoretical background, we assigned three coding techniques: open, axial, and selective coding [104], as shown in Table 2.

First, we inductively analyzed our data using a general strategy of working with the data from the ground up by searching for concepts, themes, and patterns [98]. We then assigned open codes that summarized the general concepts discussed by the interviewees. Second, axial coding was employed by grouping the concept into categories. Finally, using selective coding (in which the themes emerged), we explained the strategy used to resolve the conflict as explained by the interviewees and linked our findings to the literature. In this step, the analysis consists of identifying patterns in the data and linking the findings to our research questions.

**Table 2.** Structure of data analysis.

| Quotation | Open Code | Axial Code | Selective Code |
|---|---|---|---|
| - People who work in sales and other businesses often say that ESG is something else. It's easy to think that way, isn't it? | Departments think about ESG initiatives as separated from their work. | Different perceptions about ESG initiatives. | General ESG conflicts |
| - When you go to the field level, you're faced with a variety of issues, such as not having ideas about how to do things, not having money, not having time, etc.<br>- For example, $CO_2$ emissions need to be reduced, but I can't seem to find a way to reduce them. Or we need to replace equipment with more efficient equipment, but we can't find the budget. | Issues in meeting ESG targets at the field. | Cost, time constraints, and how they meet the ESG targets. | |
| - When it comes to whether there are systems, processes, or even standards in place to collect the proportion of women in management positions globally, in many cases there simply are none.<br>- It has become necessary to disclose non-financial ESG information in the same way as financial information, so when it comes to collecting such information, it feels like a considerable burden on the front lines. I know we have to do this in a situation where the tools are not in place and the standards are not organized, but there is conflict on how to do this, not only at our company, but at every company right now. | Lack of systematic, standardized and methodological way of collecting ESG data causing conflicts at the field. | Lack of an ESG monitoring system. | |
| - We start by sharing what society and customers expect from Ricoh. This is definitely true. Why do we do something because it's trendy when it doesn't exist? Well, that's the source of conflict, so why not? If not. If we talked carefully about the significance and purpose of that initiative in advance, I would understand what was being said, but it would increase the cost, so I just can't do it right now, and I don't want to do it. | Although they explain the meaning of their ESG target, resistance occurred in not wanting to change because of ESG principles. | Resistance | |
| - This is an example from Spain, and it is a public institution in Spain. When bidding on copiers, we evaluate various companies through bidding, and we evaluate them on a scale of 100 points. Of that 100 points, 5 or 10 points are based on ESG initiatives. What this means is that the 5 points there and the 5 points in the price are the same value. Ricoh won that bid in Spain, but according to the analysis of Ricoh's sales force in Spain, if Ricoh did not win those 5 points, if it did not win the ESG points, an additional 12% discount would be required. It is. You've already regained 5 points.<br>- We collected many examples of this and shared them within the company. We realized that this could not be done by just the sales manager. | ESG practices have become a global evaluation standard, such as part of the copier evaluation. | Convincing departments that ESG practices are a global requirement. | Strategies to overcome general ESG conflicts |
| - If we cannot get the budget, we will look for information about "government subsidies," for example, and provide it to the department, and the ESG department will help with things like applying for those subsidies. Alternatively, we provide support for such budgeting within the ESG committee. One approach to resolving conflicts is the process of working together with the department, or in other words, the ESG department, also providing support for achieving such goals.<br>- In terms of the larger direction of goals, I'd go in that direction, but as a method, I think it would be better to port it from another department, or provide support, or shift the timing a bit…the timing of the release. For those kinds of adjustments, we come in and do things like this. | Providing support in findings ways of meeting ESG goals. | ESG department support | |

**Table 2.** *Cont.*

| Quotation | Open Code | Axial Code | Selective Code |
|---|---|---|---|
| - Well, sometimes a certain kind of concrete situation arises, but perhaps it can be overcome through management judgment. That's why we set our company-wide goals at this level in each department. Then, each department, including management, should share and discuss what the goals will be.<br>- Together with financial targets, each department's ESG targets are reported to the management committee as their own business targets and approved, so it becomes something that must be achieved. | To overcome resistance, the final decision comes from management. | Management power | |
| - After all, there will be a conflict of opinion as to why we have to do this even though the cost will go up. So, is this going to be resolved? It was very difficult, but in the end the decision was made at the management level.<br>- Well, as I declared, electricity derived from renewable energy is basically a bit expensive, and regular electricity is expensive, but the customer's request, as mentioned earlier, is based on specific environmental conditions.<br>- There are five factories around the world that make multifunctional printers because environmentally friendly products and environmentally friendly manufacturing are often required in business negotiations. Thailand, China, and Japan. There are five in the world. There will be additional costs at that factory, but we have decided to convert all the electricity used to assemble the copying machine to 100% renewable energy from recycled sources based on ESG considerations. That means the electricity bill will go up. To our customers, this Ricoh multifunction device is wholly assembled using renewable energy. Since it can be promoted, well, when you compare the sales promotion, appeal effect, differentiation effect from other companies, switching to renewable energy, or additional costs, it will pay for itself well. However, for the main factory of the copier, let us recreate the assembly electricity in advance. This is an example of something that might cost more.<br>- Plastic is poured into the mold of that product, and if it was the plastic from that barge, it would flow into the mold without any problem, but if it was recycled plastic, it's like it doesn't flow properly. There were various technical issues, and it would take a lot of effort to overcome them, so it was difficult to increase the amount. | Conflict of meeting environmental targets because of high cost. | Cost of meeting environmental targets | Environmental conflict |

**Table 2.** *Cont.*

| Quotation | Open Code | Axial Code | Selective Code |
|---|---|---|---|
| - For customers, machines assembled entirely with renewable energy are more appealing.<br>- We show that the activities of the people who are doing the work will improve. And it is about getting people to understand the meaning of what we are doing.<br>- In that sense, we are actually doing ESG activities together. So, it is really important for me to convey to you that I recognize that this kind of thing exists. Thank you for always doing that together. Moreover, the results have led to good evaluations on the other side. I told you about it. Here are the external evaluation results, which are exactly the same as when I explained them to the personnel department. So, for example, there is a rating system called the Dow Jones Sustainability Index, but because of this, we are just asking everyone to give us their personal data. In fact, 3500 companies worldwide are being evaluated, and Ricoh is currently in the top 5%.<br>- It looks like the cost will go up, but since this would become a customer request in the future, we decided to do it, so we made a pretty big decision, and this new product was released. | Focus on what the market needs and predicting customers behavior, engaging workers to understand the meaning of meeting the environmental goals and the results of their contribution. | Shifting from cost-of-capital-centric to market-competitiveness-centric | Strategy to overcome environmental conflicts |
| - The moment we were asked to submit figures for the global proportion of women in management positions, we needed the cooperation of the human resources department. When I asked them to collect the ratio of women in management positions globally, why did we have to do it? | Efforts to disclose social information. | Complaints regarding disclosing of social information | Social conflict |
| - This is because if we do not disclose that information to our customers, we will not be chosen by our customers, so we have to explain this to the people in the human resources department. Alternatively, as is often the case with production factories, some factories in China or Thailand make the copiers that customers purchase. For example, is the factory's response to human rights issues correctly managed internationally? So please provide evidence. This is something I get asked a lot. Are the factories that make the copiers we purchase okay? It is asked as if to say this.<br>- In order to match that international plan, we need to ask the people at the factory to take various initiatives, but the people at the factory ask us why they have to do them. After all, this is what customers are looking for, and it is not just that they want it. Business negotiations are going on. For example, it is a business deal worth several billion yen; it is a business deal worth several tens of billions of yen. If we lose points here, we may lose to the competition. Or customers may say they can no longer buy our products unless we completely clear this. Well, that is why it has to be done. It is the expectations and demands from customers and changes in the world. If we share that kind of information not just with salespeople but with the entire group, it will lead to business growth and increased corporate value. I want them to understand that. | When asked by customers to disclose details about factories and workers conditions, it becomes part of business negotiations worth billions of Japanese yen.<br>Customer demand. | Feedback on market needs | Strategy to overcome social conflicts |

Types of conflicts include general ESG conflicts, environmental conflict, social conflict, and governance conflict. The adopted strategies to resolve the conflicts include convincing departments that ESG practices are a global requirement, ESG department support, management power, shifting from cost-of-capital-centric to market-competitiveness-centric, and feedback on market needs.

We addressed the rival explanations [98] of our case study by asking which words were used to convince business units to change to meet sustainability standards, such as ESG, SDGs, CSR, or sustainability. We found that ESG is the current term.

## 4. Results and Discussions

### 4.1. Ricoh Group ESG Strategy

The Ricoh Group started its business by providing office automation and expanded to offer office printing, office services, commercial printing, industrial printing, thermal media, and other related services. At its inception, Ricoh Group founder Kiyoshi Ichimura (1900–1968) set a sustainability vision called the principles of the spirit of three loves: love your neighbor, your country, and your work.

In 2020, Ricoh announced that it would become a digital services company by 2025 by building IT infrastructure for workplaces, such as offices, frontlines, and homes, digitizing and connecting workflows, and supporting new work practices.

The adoption of ESG in Ricoh began in 2017 by integrating ESG with its management strategy, and in 2018, it established an ESG committee.

> *The CEO chairs this committee comprised of Group Management Committee members, Audit and Supervisory Board members, and the executive officer overseeing ESG. The committee aims to enhance Group management, responding promptly and appropriately to stakeholder expectations and needs through ongoing management-level discussions of the Ricoh Group's medium- to long-term environmental, social, and governance issues [105].*

The recent ESG strategy for the 21st century medium-term business strategy was announced in 2022. The plan for 2023–2025 includes setting ESG goals to support business strategies such as digital service transformation, meeting society and customers' expectations, strengthening integration with management systems, setting 16 ESG targets and executive stock compensation systems, strengthening solutions to social issues through business, strengthening proposals to customers, and developing active advocacy activities.

Table 3 illustrates Ricoh ESG materiality and their focus domain and ESG targets [106]. A total of 16 ESG targets were set under the ESG strategy in the 21st medium-term management strategy (2023–2025).

Figure 1 shows the SDGs/ESG and the material outcomes of Ricoh. We grouped Ricoh materiality and SDG goals under the ESG dimensions based on the following definition of ESG: environmental factors in how a company performs as a steward of the natural environment. Social factors include how a company manages its relationships with its employees, suppliers, customers, and the communities in which it operates. Governance factors consist of a company's leadership, executive pay, internal controls, audits, and shareholder rights [107]. The following material outcomes overlap: creativity from work, diverse and inclusive workforce, and responsible business process. For instance, creativity from work through transitioning toward service digital business model meets two environmental goals, as it mitigates environmental impacts [108] and socially in providing customers an opportunity to work from home [109]. Additionally, some of the SDG goals meet all ESG goals, such as 17 partnerships for the goals.

One of the major changes in Ricoh's strategy is when it considers ESG goals as future finance (Figure 2), in which it envisages that pursuing ESG goals will have a positive impact on its financial performance in 3 to 10 years. It communicates the company's strategies at all firm levels and uses these to motivate employees to pursue ESG.

Different rating agencies have recognized Ricoh as a leader in ESG. MSCI identified Ricoh as a top ESG leader and a member of the leading Dow Jones Sustainability Indices. The Global 100 Index ranked Ricoh 80th in 2022 for with only four Japanese companies in the list.

**Table 3.** Ricoh material issue initiatives and ESG targets. Source: Integrated Report 2023 (P.35). Edited by the author.

| Materiality | Details | Focus Domains | ESG Targets (2023–2025) |
|---|---|---|---|
| Zero-Carbon Society | To decarbonize the entire value chain and create business opportunities by contributing to carbon neutrality. | • Environment and energy<br>• Eco-friendly MFPs<br>• Commercial and industrial printing<br>• Silicone top liner-less label and label-free printing<br>• PLAiR (material that helps reduce pollution from waste) | 1. GHG Scope 1 and 2 reduction rate (from FY2015): 50%<br>2. GHG Scope 3 reduction rate (from FY2015): 35%<br>3. Renewable energy utilization ratio for power consumption: 40%<br>4. Avoided emissions:1.4 million metric tons |
| Circular Economy | To create business opportunities by building a circular economy business model for us and our customers. | | 5. Virgin material usage ratio: 80% or less |
| Creativity from Work | To provide digital services that transform how customers work and help them with productivity improvement and value creation. | • Office services<br>• Printing industry digitalization<br>• Thermal media<br>• Industrial products<br>• Smart vision | 6. Percentage of customers considering Ricoh a digital services company: 29% |
| Community and Social Development | To contribute to the maintenance, development, and efficiency of community and social systems. We leverage our technical expertise and customer. Connections to expand areas where we provide value. | • GEMBA (maintenance and services for stores, warehouses, and other non-office sites)<br>• Biomedical<br>• Municipal solutions<br>• Educational information and communication technology solutions | 7. Number of people to whom we have contributed by improving social infrastructure: 15–20 million |
| Open Innovation | To shift from a self-sufficient approach to a new value creation process that creates businesses to quickly resolve social issues. | - | 8. Contracted joint R&D ratio: 25%<br>9. Digital services patent application ratio: 60% |
| Responsible Business Processes | To earn stakeholder trust by taking a holistic view of our supply chain and minimizing ESG risks in our business processes. | - | 10. Corporate human rights benchmark score: Information and communication technology sector leader<br>11. Compliant with NIST SP 800-171 coverage of company's core business environment: 80% or more.<br>12. Low-compliance-risk group companies: 80% or more |
| Diverse and Inclusive Workforce | To foster a corporate culture where diverse employees can demonstrate their potential and transform themselves and the company into one that is resilient to change. | - | 13. Ricoh digital skills level 2 ratings or above-rated employees (Japan): 4000<br>14. Process DX Silver Stage-certified employee ratio: 40%<br>15. Employee engagement scores<br><br>Global: 3.91<br>Japan: 3.69<br>North America: 4.18<br>Latin America: 4.14<br>Europe: 4.01<br>APAC: 4.15<br><br>16. Female-held managerial position ratio: global: 20% (Japan: 10%) |

In 2023, 37,721 employees were surveyed to understand their satisfaction with their work and how their company's SDG/ESG initiatives affected their work satisfaction. It was found that the majority, about 92%, felt engaged in solving social issues. Furthermore, 60% agreed that their efforts to resolve social issues were professionally and personally fulfilling. Furthermore, 51% agreed that the Ricoh Group's efforts to resolve social issues were professionally and personally fulfilling [106].

Setting material ESG targets created conflict at the division level. The findings below discuss the main conflicts that the Ricoh Group is facing, and the strategies used to overcome them, as shown in Table 4. We classified conflicts based on context. For instance, we refer to conflicts that occur in all ESG factors as general ESG conflicts and those that occur in each ESG factor.

*4.2. General ESG Conflicts*

At the field level, many conflicts were encountered. For example, some departments have different perceptions about ESG practices, as they consider it a separate initiative from their work and encounter difficulties meeting ESG goals, time constraints, and high costs. In some cases, resistance to comply in meeting ESG targets and a lack of systematic,

standardized and methodological way of monitoring ESG data causes conflicts in the field. The latter is the current challenge they are facing as the current development of ESG lacks standardization and systematic ways of monitoring [110].

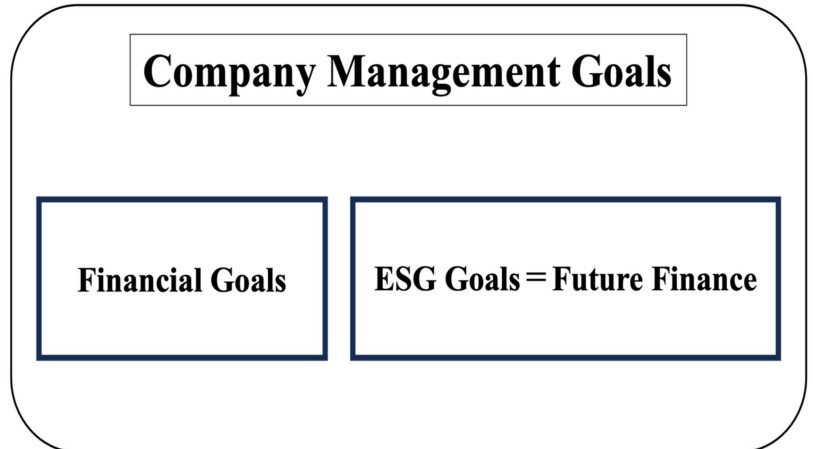

**Figure 1.** Ricoh material outcomes, ESG and SDGs. Source: Integrated report 2023. Edited by the author.

**Figure 2.** Ricoh management goals. Source: Initiatives to integrate ESG/SDGs and management strategy 2023. Edited by the author.

**Table 4.** Types of ESG conflicts and synthesis strategies for ESG conflict resolution.

| ESG Dimension | Type of Conflict | Synthesis Strategy |
|---|---|---|
| General ESG | Different perceptions about ESG practices | Convincing departments ESG practices are a global requirement |
| | Cost, time constraints and how to meet the ESG targets | ESG department support |
| | Resistance | Management power |
| Environment | Cost of meeting environmental targets | Shifting from cost-of-capital-centric to market-competitiveness-centric |
| Social | Efforts to disclosure social information | Feedback on market needs |

In the literature, ESG conflicts have been approached as a trade-off between financial and ESG performance [111], with a suggestion that the way to approach ESG is through major innovation in process, product or business model [85]. However, our findings provide a different perspective in approaching ESG conflicts through adopting a synthesis paradox strategy [30,31,42] that aims to deal with paradox by creating new perspectives and introducing new terms to resolve the paradox [42]. A synthesis strategy evolves during the integration of ESG within their operation and helps to explore new conflict-resolution strategies.

4.2.1. Convincing Departments That ESG Practices Are a Global Requirement

Ricoh convinces departments to meet the ESG targets using a synthesis strategy in which the sales department considers ESG practices a separate initiative. Ricoh shares with them and all related stakeholders the reasons for meeting the ESG standards. As Ricoh's customers are large European companies, municipalities, and government agencies, an evaluation of ESG efforts has become a part of the bidding requirements. Moreover, if it loses points, it may lose the competition.

For example, public institutions in Spain evaluated Ricoh ESG efforts as part of copier bidding. As stated by the ESG strategy department manager, the importance of meeting ESG requirements is as follows.

*Of that 100-point scale, 5 or 10 points will be evaluated regarding ESG initiatives. If we did not get those 5 points, and if they did not get ESG points, they would need a further 12% discount. It seems that it is. We have already regained 5 points, so these ESG initiatives are not just about showing off our social contribution as doing good things for the world. This is a change that customers say has become essential to gaining customer expectations and customer evaluation.*

Finally, all the cases that demonstrate the importance of meeting ESG standards in all departments are shared.

*After collecting many examples of this and sharing them within the company, we realized that this could actually be done not only by the sales manager.*

4.2.2. ESG Department Supports

The paradoxical strategy adopted is a resolution (integrative): a synthesis strategy that sets up supporting policies and cultures for integration [42].

The ESG department supports business units when needed, such as helping them find subsidies and working with them to find solutions by sharing knowledge and mutually confirming the degree of goal achievement. For example, in a factory, employees work together to discuss issues and achieve goals by sharing their know-how.

*For example, $CO_2$ emissions need to be reduced, but they cannot find a way to reduce them. Alternatively, they do not have the budget when they need to replace it with more*

*efficient equipment. If we cannot get the budget, we will look for information about "government subsidies" and provide it to the department, and the ESG department will help with things like applying for those subsidies. Alternatively, we provide support for such budgeting within the ESG committee. One approach to resolving conflicts is the process of working together with the department, or in other words, the ESG department, also providing support for achieving such goals.*

Additionally, the department made necessary adjustments such as changing the sales date when it became difficult to meet the ESG targets of a product.

*In terms of the larger direction of goals, I'd go in that direction, but as a method, I think it would be better to port it from another department, or provide support, or shift the timing a bit...the timing of the release. For those kinds of adjustments, we come in and do things like this, well, it's a formality.*

### 4.2.3. Management Power

Although they explained the meaning and importance of their ESG targets to employees, resistance occurred in not wanting to change because of ESG requirements.

*We start by sharing what society and customers expect from Ricoh. This is definitely true. Why do we do something because it's trendy when it doesn't exist? Well, that's the source of conflict, so why not? If not, if we talked carefully about the significance and purpose of that initiative in advance, I understood what was being said, but it would increase the cost, so I just can't do it right now, and I don't want to do it.*

The ESG targets were set by top-level executives and communicated as a goal that must be achieved.

*Together with financial targets, each department's ESG targets are reported to the management committee as their own business targets and approved, so it becomes something that must be achieved.*

In spite of acknowledging the importance of ESG, in some cases, workers do not want to increase their costs. In such cases, to overcome this resistance, the manager makes the final decision to resolve the tension.

*Well, sometimes a certain kind of concrete situation arises, but perhaps it can be overcome through management judgment. That's why we set our company-wide goals at this level in each department. Then, each department, including management, should share and discuss what the goals will be.*

### 4.3. Environmental Conflicts

In Ricoh's case, it assigns ESG targets to each division. However, the high cost of meeting environmental standards has resulted in internal tensions in some divisions. The integration of environmental standards has been a constant cost challenge. This indicates a performing tension that emerges from a variety of stakeholders and goals [91].

In response to this tension, Ricoh deals with the complexity of tensions with different stakeholders through communication. Resolving environmental conflicts describes how firms deal with paradoxical tensions [92]. A synthesis conflict-resolution strategies were used in setting policies and culture to support the integration [30,31,42] in shifting from cost-of-capital-centric to market-competitiveness-centric.

### Shifting from Cost-of-Capital-Centric to Market-Competitiveness-Centric

The ESG committee discussed all ESG conflicts and makes decisions to resolve them. The discussion involved setting ESG targets, breaking them into divisions, and planning to achieve them. Conflict exists in divisions when ESG targets are assigned at the division level. One of Ricoh's material ESG targets is a zero-carbon society by setting a target of a renewable energy utilization ratio (40%) for power consumption.

For instance, in 2017, the Ricoh Group joined the RE100 initiative by converting all its businesses to renewable energy. However, the cost of conversion was high, but as customers favor environmentally friendly products, they shifted their focus from cost-capital-centric to market-competitive-centric by focusing on customer needs.

> *Well, as I declared, electricity derived from renewable energy is basically a bit expensive, and regular electricity is expensive, but the customer's request, as mentioned earlier, is based on specific environmental conditions.*

After joining the RE 100 initiative, manufacturers in different countries must be convinced to convert to renewable energy sources.

> *There are five factories around the world that make multifunctional printers because environmentally friendly products and environmentally friendly manufacturing are often required in business negotiations. Thailand, China, Japan. There are five in the world. There will be additional costs at that factory, but we have decided to convert all the electricity used to assemble the copying machine to 100% renewable energy from recycled sources based on ESG considerations. That means the electricity bill will go up. To our customers, this Ricoh multifunction device is wholly assembled using renewable energy. Since it can be promoted, well, when you compare the sales promotion, appeal effect, differentiation effect from other companies, switching to renewable energy, or additional costs, it will pay for itself well. However, for the main factory of the copier, let us recreate the assembly electricity in advance. This is an example of something that might cost more.*

Despite the high cost of conversion and tensions raised in the field, Ricoh convinces its divisions that what it does is for customers, and conversion means that loyalty and the number of customers will increase accordingly.

> *For customers, machines assembled entirely with renewable energy are more appealing.*

Additionally, Ricoh communicated and provided constant feedback to all workers about its ESG achievements and ensured that they understood the value of their work.

> *If you approach things like, "Just do it because it is happening," the people in the field who are thinking, "What is the point, because the boss is saying it?" will become more and more distant, so we show that the activities of the people who are doing the work will improve. It is about getting people to understand the meaning of what we are doing.*

They also highlighted how their contributions made a difference through their selection by external evaluation, as follows.

> *In that sense, we are actually doing ESG activities together. So, it is really important for me to convey to you that I recognize that this kind of thing exists. Thank you for always doing that together. Moreover, the results have led to good evaluations on the other side. I told you about it. Here are the external evaluation results, which are exactly the same as when I explained them to the personnel department. So, for example, there is a rating system called the Dow Jones Sustainability Index, but because of this, we are just asking everyone to give us their personal data. In fact, 3500 companies worldwide are being evaluated, and Ricoh is currently in the top 5%.*

Another example of a material goal is the circular economy, the target is to increase virgin material usage ratio to 80% or less. In February 2023, Ricoh released a new copier that uses 50% recycled material (according to its own research, the amount of reused material in copiers in the market was about 20% to 30%). Everything was expensive from the beginning. The cost of virgin plastic was high, so the more they add, the higher the cost will be. In addition to the technical issue:

> *Plastic is poured into the mold of that product, and if it was the plastic from that barge, it would flow into the mold without any problem, but if it was recycled plastic, it's like it doesn't flow properly. There were various technical issues, and it would take a lot of effort to overcome them, so it was difficult to increase the amount.*

Regardless of the high cost of adopting a circular economy, Ricoh predicted that customers would request such kinds of products in the future.

> *It looks like the cost will go up, but since this would become a customer request in the future, we decided to do it, so we made a pretty big decision, and this new product was released.*

The amount of carbon footprint and $CO_2$ emissions per product reduced by 27%, and Ricoh was encouraged by customers to make more proposals for environmentally friendly products. The number of products used is increasing rapidly, thus contributing to business and sales strategies.

The environmental aspect is very much a focus among the social and governance factors in Japan [67]. However, limited discussion of the conflict exists. Additionally, the discussion of environmental matters were limited to climate change and short and long tensions (intemporal tensions) [29]. Moreover, the circular economy is viewed as a condition for sustainability, a beneficial relation, or a trade-off [112]. However, our findings provide a different approach in viewing meeting the circular economy targets as shifting from cost-of-capital-centric to market-competitiveness-centric.

### *4.4. Social Conflicts*

In Ricoh's case, social conflicts occurred when the company requested that the human resources department disclose the percentage of female managers locally, globally, and at all branches. Additionally, it wanted to increase the number of female managers at the divisional level and ensure human rights at the factories. Furthermore, feedback on market requirements is used to resolve conflicts.

Feedback on Market Needs

Two material ESG goals were met with conflicts. First, Regarding responsible business processes, Ricoh's goal is to be a leader in the sector in corporate human rights benchmark score. Second, with a diverse and inclusive workforce with female-held managerial positions, we found that a stakeholder paradox exists when integrating ESG goals [113]. This tension is addressed using a synthesis strategy [30,31,42] that focuses on customer demands. For instance, the human resources department complained about the effort to collect and disclose information on human rights in factories and the percentage of female managers. Ricoh convinced them of the value of disclosing to customers when they asked about human rights at the factories, and that if they did not disclose this information, they might lose their customers.

> *This is because if we do not disclose that information to our customers, we will not be chosen by our customers, so we have to explain this to the people in the human resources department. Alternatively, as is often the case with production factories, some factories in China or Thailand make the copiers that customers purchase. For example, is the factory's response to human rights issues correctly managed internationally? So please provide evidence. This is something I get asked a lot. Are the factories that make the copiers we purchase okay? It is asked as if to say this.*

Ricoh also informed them that they had been used in business negotiations for large-budget amounts.

> *In order to match that international plan, we need to ask the people at the factory to take various initiatives, but the people at the factory ask us why they have to do them. After all, this is what customers are looking for, and it is not just that they want it. Business negotiations are going on, for example, it is a business deal worth several billion yen; it is a business deal worth several tens of billions of yen. If we lose points here, we may lose to the competition. Or customers may say they can no longer buy our products unless we completely clear this. Well, that is why it has to be done. It is the expectations and demands from customers and changes in the world. If we share that kind of information*

*not just with salespeople but with the entire group, it will lead to business growth and increased corporate value. I want them to understand that.*

In the past, most managers were men, and there was conflict in the human resources department over what the target ratio of female managers should. Since the target number was high, the department started to break this down into manageable targets to be achieved at certain time points. Discussions were held at the human resources department about what target level to set should take place.

The literature on solving social conflicts using the paradox perspective is limited with regard to social justice (providing living wage to workers), financial performance [44], and pursuing social and financial goals [89].

**5. Conclusions**

This study examined how a top ESG leader in the information and technology sector, the Ricoh Group, managed ESG conflict in the process of ESG integration. Although the literature provided a rich discussion on resolving paradox tensions and conflict, our findings extend the literature by providing different strategies that have not been discussed yet. Additionally, the literature has focused on examining managers' behavior with regard to sustainability and tensions [42–47], while in our study, we examined the tensions that occur at the field level.

We classified ESG conflicts according to their type. For instance, general ESG conflicts involve different perceptions of ESG principles, meeting ESG goals, time constraints, high costs, resistance, and the lack of an ESG monitoring system. Environmental conflicts include the high costs of adopting environmental standards, and social conflicts in ensuring human rights at the factories and an increase in the percentage of female managers.

We found that Ricoh adopts a synthesis strategy [30,31,42] of ESG conflict resolution for general ESG practices in terms of convincing employees that ESG as a global requirement and providing support and management power. Environmental conflicts are resolved by shifting the perspective from cost-of-capital-centric to market-competitiveness-centric, and resolving social conflicts through feedback on market needs.

This study is the first to examine ESG conflicts in the field and the strategies adopted to resolve them. We contribute to the literature by highlighting ESG conflicts and the strategies adopted to resolve these conflicts [30], and provide an understanding of tension from the field. Additionally, we highlighted the current challenge that causes conflicts is the absence of a well-developed ESG monitoring system. Moreover, we state that approaching ESG pursuits as future finance may support the integration process as it motivates employees to work toward meeting ESG targets.

We provide the following implications for stakeholders. First, the government should provide guidelines for ESG conflicts and resolution strategies to overcome them. Second, policymakers should provide education on ESG conflict management and the necessary support for firms to address conflicts through constant feedback. Finally, we suggest using Ricoh's ESG conflict-resolution strategy to understand the types of conflicts and strategies used.

Although a case study helps to examine real phenomena, it has been criticized for difficulty in generalization and second-rigor methodology [98]; however, while acknowledging these limitations, regarding the first limitation, we interviewed the ESG department, which provided sufficient insights to address our research questions. Additionally, we addressed the limitations of attempting to make the research procedure transparent.

We offer the following seven suggestions for future research. First, collaboration is required between academia and professional international organizations such as the Global Reporting Initiative (GRI) and Sustainability Accounting Standards Board (SASB) to develop a systematic ESG monitoring system based on the firm sector and context. Second, more case studies are needed to test and extend our findings to different sectors, firm sizes, countries, and contexts, such as countries that mandate ESG and voluntary reporting. Third, comparing successful and failed ESG conflict-resolution strategies is

needed to identify factors contributing to successful conflict and tension resolution. Fourth, we state the need to examine governance conflicts and strategies used to overcome them. Fifth, classifying how the paradox strategy differs from other strategies in terms of ESG conflict resolution is necessary. Six, more studies are needed to explore different types of ESG conflict management from different corporate sustainability perspectives. Finally, more studies to examine the influence of the culture on conflict-resolution strategies during integration of ESG practices are needed.

**Author Contributions:** Conceptualization, A.A., J.K. and O.A.; methodology, A.A. and J.K.; software, A.A.; validation, O.A.; formal analysis, A.A.; writing—original draft preparation, A.A.; writing—review and editing, A.A, J.K., O.A. and H.C.G.; supervision, A.A. All authors have read and agreed to the published version of the manuscript.

**Funding:** This research received no external funding.

**Institutional Review Board Statement:** Not applicable.

**Informed Consent Statement:** Informed consent was obtained from all subjects involved in the study.

**Data Availability Statement:** Data are contained within the article.

**Acknowledgments:** Special acknowledgement to the Ricoh Group for the cooperation throughout the research and data collection stage.

**Conflicts of Interest:** The authors declare no conflicts of interest.

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
