# Peer review of "How to Manage Conflicts in the Process of ESG Integration? A Case of a Japanese Firm"

_sustainability, doi:10.3390/su16083391_

Round 1
Reviewer 1 Report (Previous Reviewer 2)
Comments and Suggestions for Authors
Congratulations to the authors for the significant improvement and the great effort made to craft an exciting original research paper.
Author Response
Please see the attachment

Reviewer 2 Report (Previous Reviewer 3)
Comments and Suggestions for Authors
-
The study appears to have been conducted ethically, with informed consent from all participants. There's no indication of a conflict of interest, but it may be beneficial for the journal to ensure that all ethical considerations, especially those related to corporate case studies, are thoroughly addressed.
-
There is no evident concern regarding plagiarism or inappropriate self-citations. However, a thorough check to confirm the originality of the manuscript and the appropriateness of citations would be prudent.
-
The paper does not raise additional ethical concerns. The research process seems transparent, and the Ricoh Group's involvement and the non-financial nature of the study's support are clearly stated.
The review suggests several areas for improvement to enhance the paper's clarity, depth, and relevance:
-
Enhance Contextualisation: Deepen the discussion on how the findings align with or diverge from existing literature, especially regarding the application and effectiveness of paradox theory in ESG integration. Highlighting the specificities of the Japanese corporate culture or the tech industry context could provide richer insights.
-
Expand Literature Review: To enrich the theoretical underpinning of the study, incorporate a broader range of recent and diverse sources critiquing or expanding on paradox theory within the context of ESG.
-
Methodology Transparency: Offer more details about the interviewee selection process and the formulation of interview questions. Clarify how data analysis (beyond coding) led to theme derivation, enhancing the study's validity.
-
Alternative Interpretations: Explore and discuss alternative interpretations of the data, including comparisons with firms outside the tech sector or in different geographical regions, to underscore the uniqueness or universality of the findings.
-
Visual Aids: Use charts or models to illustrate the synthesis strategy for ESG conflict resolution, making the findings more accessible and engaging.
-
Update References: To maintain the paper's relevance, ensure all references are current and reflect the latest research in ESG integration and conflict management.
-
Support for Conclusions: Strengthen the conclusions by discussing future directions for ESG conflict management, considering emerging sustainability and corporate governance trends.
-
Quality of English: Minor edits for clarity and flow, with consistent terminology use related to ESG and conflict resolution throughout the paper, would improve readability.
Author Response
Please see the attachment

Reviewer 3 Report (Previous Reviewer 4)
Comments and Suggestions for Authors
Dear Aurthors,
The presented study concerns conflict management in corporations and overcoming resistance to change. It describes such activities and discusses the organization's management policies, including changes to company policy and ways of communicating the need for change.
What is new is the presentation of how ESG principles are implemented.
The article provides extensive explanations of the attitudes, motivations, and actions taken in various company departments. The literature was selected appropriately and sufficiently.
The main comment regarding the work is that the "conclusions" section continues the discussion of the results. You might consider moving some of this description to some new short section 4.5.
It is worth considering whether the abstract should also provide more direct information about what actions the authors found to have the best impact on the change process and what actions will influence the company's economic results in the future.
I have no comments on the quality of text formatting and bibliography list.
Author Response
Please see the attachment

Reviewer 4 Report (Previous Reviewer 7)
Comments and Suggestions for Authors
The manuscript was significantly impoved. The originality and the engagement eith sources as well as recent scholarship is high. The overal metric of the manuscript is average.
Author Response
Please see the attachment

This manuscript is a resubmission of an earlier submission. The following is a list of the peer review reports and author responses from that submission.
Round 1
Reviewer 1 Report
Comments and Suggestions for Authors
First of all, thanks to the authors for considering this journal for their work.
The structure of the paper is clear. The objectives and contributions are well defined. Also the literature review is adequate and relevant. However, the justification of why the authors only choose one case study does not seem sufficient, as this clearly makes it much more difficult to generalise the results and apply the proposed solutions to any entity, where it is common for each one to present a different context or particular situation.
Similarly, the questions chosen and possible solutions proposed are sometimes difficult to justify why these and not others. Perhaps a greater number of cases, or previous studies that support the authors' results, would reinforce and give greater validity to the proposal. There is little basis/justification for the method and results/conclusions reached.
Reviewer 2 Report
Comments and Suggestions for Authors
The paper you have written is original and has implications for theory, policy, and managers. I would like to draw your attention to a few minor observed errors. On page 3, lines 100 - 106, correct the sources. You cannot use endnotes and parentheses- the author citation system at the same time. This suggestion also should be applied to the rest of the paper where you used parentheses - the author citation system (lines 116, 124, 126, 134, 136, 142, 143, 147, 152, etc.). Page 7, line 319: 4.3.1 Shifting from cost-of-capital-centric to market-competitiveness-centric. You should remove a dot at the end of 4.3.1. subheading. Same for 4.3.2 Convincing departments that ESG is a global requirement, page 8, line 369.
Page 9, line 396 4.4.1 feedback on market needs. Instead, it should be written in the following way: 4.4.1 Feedback on market needs.
I suggest that in Conclusions, separate in the form of subtitles and explain in more details:
-Theoretical contributions
- Policy and managerial implications
- Limitations and suggestions for future research.
I hope these suggestions will be helpful.
Comments on the Quality of English Language
I have no additional comments.
Reviewer 3 Report
Comments and Suggestions for Authors
I would suggest a major revision for the following reasons:
-
- The abstract and introduction provide a decent overview, but the linkage between the presented research and existing literature could be strengthened. The study's positioning within the broader context of ESG research needs more clarity.
-
- While the paper cites a range of sources, there seems to be an over-reliance on certain references. A more diverse range of sources could enrich the theoretical foundation and relevance.
-
- The methodology section lacks detail, particularly in the justification of the case study approach and the selection of the Ricoh Group. More transparency in methodological choices would enhance reliability.
-
- The discussion section connects findings to existing literature but lacks critical analysis. A more nuanced discussion about how the findings advance or challenge current understanding would be beneficial.
-
- The results are presented with clarity, but the integration of these results into the broader narrative of the paper could be enhanced. Visual aids or clearer thematic categorisation could help.
-
- The paper is adequately referenced, but a more thorough engagement with recent studies and critical perspectives could strengthen its scholarly depth.
-
- The conclusions draw on the results but could benefit from deeper analysis. The linkage between the specific findings and broader implications needs more explicit articulation.
Ratings:
- Originality: Average - The study's focus on ESG conflicts offers a novel perspective, but more innovative approaches in methodology or analysis could be explored.
- Contribution to Scholarship: Average - It contributes to existing literature but lacks a transformative or significantly novel insight.
- Quality of Structure and Clarity: Low - The paper's overall structure needs refinement for better flow and coherence.
- Logical Coherence/Strength of Argument/Academic Soundness: Average - Arguments are logical but lack depth in places, affecting the paper's academic robustness.
- Engagement with Sources and Recent Scholarship: Average - Engagement is present but needs to be more critical and encompassing of recent debates.
- Overall Merit: Average - The paper holds merit in its focus and findings but requires significant improvements to reach its full potential.
Reviewer 4 Report
Comments and Suggestions for Authors
Dear Authors,
the study concerns the current issue of introducing and requiring ESG in supply chains.
This is of great importance for maintaining the competitiveness of a given business in an environment in which declaring the introduction of ESG goals has become important.
It is also important that an attempt has been made to show how corporations deal with resistance to change, although it appears that the techniques used are the same as in the case of implementing other strategies.
The main reservations regarding the report are related to the insufficient connection between the description and the actual ESG goals of the company being examined and the lack of showing the assumed level of indicators.
Another weakness that I notice is the lack of assessment of the justification for introducing ESG, linking such a policy with existing activities related to environmental protection (energy consumption indicators, eco-friendliness of used operating materials, product design with recycling in mind, etc.) or the principles of employee treatment and compliance with employee rights. Is ESG something significantly new or just another extension of existing activities in a "new package?" What percentage of the global market requires an ESG declaration? Are there any reference levels? The literature review should be extended to include such aspects, and the company description should include information about the ESG goals and the most important target indicators.
General remarks:
You need to specify the ESG goals of the company being audited. What are the target indicators, e.g., CO2 emissions, reducing plastic consumption, or what goals are set?
Specify more specifically what "ESG Factors" means; it can be from the literature (companies indicate different goals, but they are repeatable) or from the documents of the audited company Ricoh.
In section 1, you should declare how the author understands "corporate sustainability" concerning ESG. Are these the same concepts? How does the author put it?
In the literature review section (2), it is worth presenting whether and what assessments are made of ESG implementation requirements. Does it mainly lead to achieving business goals (maintaining a position on the market), or are ecological and social goals mainly achieved?
In the literature review, there should also be added reasons for the resistance/slow/partial implementation of ESG in some countries and regions of the world. Whether and why end buyers want to pay extra for ESG declarations.
Whether and why the implementation of ESG leads to a conflict between the cost incurred and the value currently obtained.
Details:
[line 50-51] It was indicated that the introduction of ESG is to facilitate financing in the future. Is such discrimination by markets justified? Should actual achievements (e.g. CO2 emissions/energy consumption per manufactured product) be taken into account?
[line 52-54] Constant pursuit of the goal and reporting on progress. Is this still being written about? In the part of the study regarding the literature review and company description, several examples of goals to be achieved and the level of indicators should be provided (e.g. CO2 emission reduction by 23.5%).
[line 85-86] What is the cause and effect. Are better-performing companies more willing to implement ESG, or does ESG implementation enable weaker companies to gain advantages over others?
[line 90-91] It is worth mentioning at least a few of the most important, repeatable "ESG Factors" here. What are the most common in enterprises.
Section 2.2. - there is too much about tensions and conflicts. Avoid repetitions - e.g. in lines 102-117.
[line 123-130] It is better to explain the relationship between CSR and ESG. Is it possible to join ESG?
[line 131 onwards] The content is similar to what is described in lines 98-122. Maybe this should be combined and reworded.
[line 142] It is necessary to indicate whether the approach according to HAHN ET AL. (2015) is adopted in this work as the basis of the methodology.
[line 145] Explain what "Global Standards" were considered. Whether the ESG concept itself or some specific solutions. Who develops these "global standards"?
[line 151-166] The description concerns social engineering in the company. These are well-known issues. Is it necessary for this job?
[line 185] Where do "normative" aspects come from? Are there any classically established and accepted standards (similar to GHG emissions per production unit should be reduced)?
[line 180-206] I propose to reword this fragment. Simplify the description of these strategies because there are repetitions. For example, what does Ferrel state? How does ESG solve financial conflicts? Is this some miracle cure for conflict?
[line 216] Unclear. Does a large number of interactions mean that the company is large, or does a large company have many interactions?
[line 246-248] Do the authors notice any differences between the Ichimura approach and the ESG approach? Can both be the same or is it some extension of the idea?
[line 248-250] Is this information necessary?
[line 261] State what these 16 ESG goals are and what the target indicators are.
[line 282] Do the authors mean "each ESG factor" or "individual ESG Factors".
[line 284] indicates that SDG/ESG targets were examined. Do the authors equate SGD goals with ESG goals (worth discussing in more detail in the literature review sections). Is Ricoh include their SDGs into ESG. It has to be clearly described.
[ad line 299-306] Does the company have any available indicator, e.g. total CO2 emissions in the production, use and waste phases? it is possible to close CO2 emissions.
[line 313-314] Sentence without use - too general. What was served there K. Van Bommel?
[section 4.3.2] It appears that sections 4.3.1. and 4.3.2 be combined. For example, in line 327 there is the customer's needs, and in lines 376 and next is description of such special requirements.
[line 400] Do you mean women - employers, or women - employees. What level of female participation among employees is required (is there any expected level of indicator, e.g. 75% is better than 55%).
[section 4.4.1] What does "human rights" mean in Ricoh factories. it should be described when Ricohs' 16 ESG goals will be presented in text. Without ESG, was human rights in factories not be respected?
[line 450-455] Do you mean the government of the country. Can ESG be recognized on a national scale?
[line 461 et seq.] You need to summarize the results in relation to ESG standards (or lack thereof), required levels of ESG goals, determining who is the beneficiary of ESG implementation (company, society, climate, planet).
Conclusion: The study is quite interesting and concerns a current issue. I believe that it should be partially supplemented with a review of the literature and an indication of the company's ESG goals and their achievement. Some fragments contain numerous repetitions and require rewording. The Conclusions section is worth referring to the most important issues regarding ESG in the company. It should also be emphasized that this is about reducing resistance to changes in connection with the implementation of ESG, and less about the description of the ESG goals and factors themselves.
Other comments. [line 98-206] When citing in text, you use a referencing style other than the required one. Is this a leftover from submitting a manuscript to another journal? This needs to be improved.
End
Reviewer 5 Report
Comments and Suggestions for Authors
This paper focuses on a critical topic in the literature. Following improvements can be very helpful to improve the quality of the papers.
1) I suggest the authors to improve the introduction section. Authors should better highlight the objective of their work and to what extent it contributes to close a gap in the existing literature and/or practice.
2) Which is the innovative value of the contribution proposed by the authors?
3) The authors must define motivations and contributions of this paper.
4) You should provide more recent references published in last two-three years in the Literature review. Remove references published before 2018.
5) The authors will have to demonstrate the impact and insights of the research.
6) The authors need to clearly provide several solid future research directions.
7) Clearly state your unique research contributions in the conclusion section.
8) Add limitations of the model.
Reviewer 6 Report
Comments and Suggestions for Authors
The authors proposed an interesting article in the field of conflict management, expanding the existing theory in the context of ESG.
The proposed methodology is strongly determined by the fact that the research is based on a case study. An undoubted disadvantage of the survey solution may be the fact that respondents may feel pressure to provide "correct" answers. However, it is difficult to indicate alternatives considering that the research takes place in the environment of a specific company.
I believe that in this context the study has an irremediable flaw, which, however, does not rule out the possibility of using its results on a broader research scale. However, the possibilities of drawing conclusions about the proposed solutions have a limited ability to translate them into the broad context of reasoning about the examined problem.
I would suggest emphasizing the limitations of inferences in the study's conclusions.
Reviewer 7 Report
Comments and Suggestions for Authors
I recommend adding the culture aspect to subchapter 2.1, because culture has a great impact on the social aspect. The case study presented in the article was based on 3 interviews realised in Ricoh Group, so the conclusions and results of the research cannot be used widely, only in relation to the investigated company. In my opinion, this is the weakest point of the article.